# A Near Linear Query Lower Bound for Submodular Maximization

**Binghui Peng** [1]   **Aviad Rubinstein** [2]

## Abstract

We revisit the problem of selecting $k$-out-of-$n$ elements with the goal of optimizing an objective function, and ask whether it can be solved approximately with *sublinear query complexity*. For objective functions that are *monotone submodular*, [Li, Feldman, Kazemi, Karbasi, NeurIPS'22; Kuhnle, AISTATS'21] gave an $\Omega(n/k)$ query lower bound for approximating to within any constant factor. We strengthen their lower bound to a nearly tight $\widetilde{\Omega}(n)$. This lower bound holds even for estimating the value of the optimal subset. When the objective function is *additive*, we prove that *finding* an approximately optimal subset still requires near-linear query complexity, but we can *estimate* the value of the optimal subset in $\widetilde{O}(n/k)$ queries, and that this is tight up to polylog factors.

## 1. Introduction

We consider the problem of selecting the "best" $k$-out-of-$n$ elements, e.g. selecting $k$ locations to place sensors, selecting $k$ features to include in data analysis, or selecting $k$ samples for training a model. We are particularly interested in the case where:

**Expensive query access** We are not given an explicit function to compute what makes one subset "better" than another: rather we have expensive query access to an oracle that evaluates different candidate subsets (e.g. estimating the quality of prediction from a subset of features or samples by training a smaller model on them).

**Sublinear query complexity** Motivated by the expensive query constraint and large problem sizes in machine learning applications, we ask whether it is possible to obtain approximately optimal subsets with query complexity that

[1]Department of Computer Science, Stanford University, United States [2]Department of Computer Science, Stanford University, United States. Correspondence to: Binghui Peng <binghuip@stanford.edu>, Aviad Rubinstein <aviad@cs.stanford.edu>.

*Proceedings of the 42nd International Conference on Machine Learning*, Vancouver, Canada. PMLR 267, 2025. Copyright 2025 by the author(s).

scales sublinearly with $n$.

In full generality, it is clearly hopeless to find an approximately optimal $k$-subset without querying all $\binom{n}{k}$ subsets (not to mention sublinear query complexity), so as is common in the literature we restrict our attention to *monotone submodular* objective functions, i.e. we assume that the marginal contribution from each additional element is diminishing, yet it is always non-negative (see Section 2 for a formal definition). Since our result for worst-case functions in this class is quite negative, we also consider the most important special case: (monotone) *additive* objective functions, i.e. $f(S) = \sum_{i \in S} w_i$ for unknown $w_i \geq 0$.

Monotone submodular maximization has important applications in machine learning (see e.g. the recent survey of Bilmes (2022) and references therein), and the problem sizes in those machine learning applications increases rapidly. This inspired over the past decade an extensive effort in obtaining sublinear time algorithms for approximate submodular maximization by parallelization (Balkanski et al., 2016; Balkanski & Singer, 2018; Balkanski et al., 2018; 2019; Chekuri & Quanrud, 2019a;b; Chen et al., 2019; Ene & Nguyen, 2019; Fahrbach et al., 2019; Kazemi et al., 2019a; Breuer et al., 2020; Ene & Nguyen, 2020; Balkanski et al., 2022a;b). In contrast, in this work we look for algorithms whose total work (specifically, total query complexity) is sublinear. Indeed, this is motivated in part by the increasing monetary and environmental cost of total work of state-of-the-art machine learning algorithms that already achieve significant parallelization. Beyond these important connections to applications in machine learning, submodular maximization itself is generally a fundamental problem in optimization.

The question of sublinear-query algorithms for submodular maximization was considered by Li et al. (2022); Kuhnle (2021), who showed that any constant factor approximation algorithm requires query complexity $\Omega(n/k)$; Li et al. (2022) additionally proved a lower bound of $\Omega(\frac{n}{\log(n)})$ for the special case of $k = \Theta(n)$. Hence, for the special case of $k = \Theta(1)$ and $k = \Theta(n)$, previous work rules out truly sublinear query complexity algorithms.

Our first result strengthens this impossibility result by ruling out approximate submodular maximization in sublinear query complexity for any $k \ll n$, even if we're just inter-

ested in estimating the value of the optimal subset:

**Theorem** (Submodular: informal version of Theorem 3.9)**.**
*With a monotone submodular objective function and for any $k = o(n)$, no $\widetilde{O}(n)$-query complexity algorithm can approximate the value of the optimal $k$-subset to within any constant factor.*

We note for typical applications of submodular maximization (e.g. dataset selection – selecting a small dataset to train a model; influence maximization – selecting a small subset of influential users), the subset size $k$ is neither linear in the entire dataset, nor a small constant that can be ignored. Comparing with previous work (Li et al., 2022; Kuhnle, 2021), we obtain improved (and optimal) lower bound for the critical regime of $\text{polylog}(n) \leq k \leq \frac{n}{\text{polylog}(n)}$. Our lower bound rules out sublinear query algorithms over all possible regimes of $k$ and fully resolves the query complexity of monotone submodular maximization (up to polylog factor), a fundamental question in optimization theory.

Given this sweeping negative result, we turn our attention to characterizing the complexity of selecting an approximately optimal $k$-subset with an additive objective function. Here, we have a mix of negative/positive results, where the key difference depends on whether we just want to estimate the value of the optimal subset, or actually find it:

**Theorem** (Additive: Informal version of Theorems 3.7, 4.1, 4.7)**.**

*With a monotone additive objective function and for any $k = o(n)$,*

- *No $\widetilde{O}(n)$-query complexity algorithm can* find *an approximately optimal $k$-subset, for any constant approximation factor (Theorem 3.7).*

- *For any constant $\epsilon > 0$, there exists an $(1 \pm \epsilon)$-approximation algorithm for* estimating *the value of the optimal $k$-subset using only $\widetilde{O}(n/k)$ queries (Theorem 4.1).*

  - *Furthermore, this query complexity is nearly tight for any algorithm obtaining any constant factor approximation (Theorem 4.7).*

Our algorithm uses sublinear queries and estimates the *value* of the optimal $k$-subset, this is a standard goal in the field of sublinear algorithms, see (Chen et al., 2022; Charikar et al., 2023; Behnezhad, 2023; Behnezhad et al., 2023; Bhattacharya et al., 2024) and reference there in. To motivate this, consider a scenario where one needs to select a subset from a large dataset under a budget constraint. Our algorithm can first be used to assess whether the dataset is sufficiently valuable – that is, whether it contains elements with large values. If the dataset meets this criterion, one

can proceed with further analysis or selection; otherwise, unnecessary efforts can be avoided.

**Query complexity vs. Runtime** The focus of this paper is on the query complexity, which is standard in the literature. Our algorithm uses sublinear queries (i.e., $\widetilde{O}(n/k)$) and linear computation time (i.e. $\widetilde{O}(n)$), both of which are optimal[1]. The motivation for studying query complexity arises from practical concerns in training large models using only a subset of the data. Even under the simplest assumption—an additive function – we lack explicit input specifying each data point's marginal value. Instead, we estimate the value of a subset by, for example, training a smaller model from scratch. Related applications include data valuation (Ilyas et al., 2022), where a query to subset $S$ corresponds to training a neural network over set $[n] \setminus S$; influence maximization (Kempe et al., 2003), where a query to subset $S$ corresponds to a simulation or a social experiment using $S$ as the seed set. In all these settings, queries are significantly more expensive than individual computational steps: while linear computation time (e.g., loading or processing the full dataset) is feasible, making a linear number of queries (e.g., training $n$ different models) is impractical.

**Additional related work** The query complexity of monotone submodular maximization has been studied in the literature. The greedy algorithm (Fisher et al., 1978) finds an $(1 - 1/e)$-approximate solution using $O(nk)$ queries. The query complexity can be improved, the stochastic greedy algorithm (Mirzasoleiman et al., 2015) makes $O(n \log(1/\epsilon))$ queries and return an $(1-1/e-\epsilon)$ approximate solution; (Li et al., 2022) gives a deterministic algorithm using $O(n/\epsilon)$ queries. Despite of extensive research, the only query lower bound known is $\Omega(n/k)$ from the recent work of (Li et al., 2022; Kuhnle, 2021).

In addition to previously mentioned work on parallel algorithms for submodular maximization, our work is also related to algorithms for submodular maximization in other sublinear models such as dynamic (where the update time needs to be sublinear) (Chen & Peng, 2022; Peng, 2021; Peng & Rubinstein, 2023; Lattanzi et al., 2020; Monemizadeh, 2020; Dütting et al., 2023; Agarwal & Balkanski, 2023; Banihashem et al., 2024; 2023) and streaming (where the space needs to be sublinear) (Badanidiyuru et al., 2014; Feldman et al., 2023; Chakrabarti & Kale, 2015; Chekuri et al., 2015; Feldman et al., 2021; Huang et al., 2021; Liu et al., 2021; Shadravan, 2020; Norouzi-Fard et al., 2018; Alaluf & Feldman, 2019; Agrawal et al., 2018; Kazemi et al.,

---

[1]It is easy to see that linear computation time is necessary: Imagine there is one (unknown) element that has huge value comparing to all other elements, an algorithm needs to at least load that element to estimate its value, which takes $\Omega(n)$ time in expectation.

2019b; Huang et al., 2022; Feldman et al., 2018; Alaluf et al., 2022; McGregor & Vu, 2019; Indyk & Vakilian, 2019).

### 1.1. Technique Overview

**The linear lower bound for submodular maximization**
The key idea driving our approach for proving lower bounds is to relate the *query complexity of submodular maximization* to the *communication complexity of distributed set detection problem*.

We first describe the communication problem of distributed set detection task. The distributed set detection is a multi-party communication problem, where each party observes the outcome of $n$ coins. Among these $n$ coins, $k$ coins are fair and have mean $1/2$, while the rest $n-k$ coins are biased and have small mean vallue. The goal of these parties is to collectively identify a small fraction (e.g. $0.1k$) of fair coins. We prove that the distributed set detection requires $\widetilde{\Omega}(n)$ communication cost using the distributed strong data process inequality (SDPI) and a direct-sum argument.

Our key observation is that the linear communication lower bound for distributed set detection yields a linear query lower bound of submodular maximization. To this end, consider a linear function whose weight on the $i$-th element equals the summation of the outcome of the $i$-th coin. The optimal $k$-subset is exactly the $k$ fair coins, given the number of parties is roughly $\Theta(\log(n))$. The crucial observation is one could simulate the query algorithm in the communication model, with only $\mathrm{polylog}(n)$ overheads: If the query algorithm asks for the value of $f(S)$ for some set $S$, then all parties just locally compute the summation of coins in $S$ and broadcast their results, it takes at most $O((\log(n))^2)$ communication bits ($O(\log(n))$ parties and $\log(n)$ bits per party). This proves that *finding the optimal $k$-subset* requires $\widetilde{\Omega}(n)$ queries due to the $\widetilde{\Omega}(n)$ communication lower bound of distributed set detection.

When the goal is to *estimate the value of the optimal $k$-subset*, the above hard instance fails because the query algorithm could easily find one fair coin using $\widetilde{O}(n/k)$ queries. To this end, we construct a monotone submodular function by applying two levels of truncation to the above linear function. Roughly speaking, the optimal $k$-subset still corresponds to the $k$ fair coins in distributed set detection, but the two-level truncation over the linear function (see Eq. (2)(3)) masks off useful information on large sets and requires the detection of a non-trivial fraction of the fair coins (see Section 3.2 for detailed description).

*Remark* 1.1 (Comparison with previous work (Li et al., 2022; Kuhnle, 2021)). The previous work (Li et al., 2022; Kuhnle, 2021) prove a lower bound of $\widetilde{\Omega}(n/k)$ and (Li et al., 2022) additionally has a lower bound of $\widetilde{\Omega}(n)$ when $k = \Theta(n)$. Our approach yields significantly stronger lower bound, e.g. when $k = \sqrt{n}$, our lower bound (i.e., $\widetilde{\Omega}(n)$) is quadratically

better than (Li et al., 2022; Kuhnle, 2021) (i.e., $\widetilde{\Omega}(\sqrt{n})$). From a technical point of view, all previous lower bounds are proved using counting arguments. Our technique is completely different from them, it is based on novel ideas, including (1) the query to communication reduction, (2) a communication lower bound using information complexity and distributed data-processing inequality, and (3) a new construction of the hard submodular function with two-level truncation.

**Sublinear algorithm for linear function**   Our algorithm is a multi-scale combination of two base subroutines. The first subroutine is to randomly select a set of size $n/m$ and estimate the value of each individual element. This subroutine yields a good estimate on the top $m$-th quantile of the ground set, but fails to give an accurate estimation on top $o(m)$ elements (to see this, imagine there are $o(m)$ elements that have super large value, then one can observe their value only after sampling $\gg n/m$ elements). The second subroutine is to partition the ground set into $m$ subsets and estimate the value of each subset. Intuitively, this subroutine alleviates the weakness of the first subroutine – if there are $o(m)$ elements that have super large value, then most of them would fall into different subsets and we can have a good sense of their value after querying the value of $m$ subsets. Nevertheless, the second subroutine could fail if the top $o(m)$ elements are not significantly larger than rest elements (say they have value 2 and the rest have value 0 or 1, then querying the value of $m$ subsets gives negligible hints on the value of top $o(m)$ elements).

Intuitively, both subroutine have their own weakness, but they are somewhat complementary to each other. We wish to combine them in a careful manner to get an optimal sublinear algorithm. The real situation gets complicated due to the possible intricate choice of the top $k$ elements, so we only sketch our final solution here. Indeed, we compose the two base subroutines in multiple scales. Let $k_r = (1 + \epsilon)^r$, at each level $r$, the algorithm randomly partitions the ground set into $nk_r/k$ subsets and estimates the $k_r$-th quantile of these subsets (hence each level takes $\widetilde{O}(n/k)$ queries). Roughly speaking, we expect the top $k_r$-th subsets to be as valuable as the top $k_r$-th element + an average bucket. Our final estimate is a weighted average over the output at each scale. See Section 4 for details.

**Organization of the paper**   We describe notations and models in Section 2. The lower bound for submodular maximization is presented in Section 3, and the algorithm for additive function is in Section 4. Due to space constraints, we defer detailed proof to the appendix.

## 2. Preliminary

**Notation** We write $[n] = \{1, 2, \ldots, n\}$. For random variables $X, Y$, we use $I(X;Y)$ to denote the mutual information of $X, Y$, $h(X;Y)$ to denote the Hellinger distance, $\mathsf{TV}(X;Y)$ to denote the total variation distance. For any value $p \in [0, 1]$, let $B_p$ the Bernoulli distributions with mean $p$.

**Submodular maximization** Let $f : \{0, 1\}^n \to \mathbb{R}^+$ be a nonnegative set function. For any sets $A, B \subseteq [n]$, let $f_A(B) := f(A \cup B) - f(A)$ be the marginal value of a set $B$ w.r.t. a set $A$. The function is monotone if $f(B) \geq f(A)$ for any $A \subseteq B$. The function is said to be submodular if $f_A(u) \geq f_B(u)$ holds for every sets $A \subseteq B \subseteq [n]$ and every element $u \in [n] \setminus B$. In a constrained monotone submodular maximization problem, there is a budget $k \in [n]$ and the goal is find a subset $S \subseteq [n]$ of size at most $k$ that (approximately) maximizes the function value, i.e., $\max_{S \subseteq [n], |S| \leq k} f(S)$. The problem is studied in the query oracle model, where each time the algorithm submit a query $V \subseteq [n]$ and the oracle returns the function value $f(V)$.

Let $S^* := \operatorname{argmax}_{S \subseteq [n], |S| \leq k} f(S)$ be the optimum solution set (breaking ties arbitrarily). In the search problem, the goal is to find an (approximately) optimal solution, and we say the algorithm finds an $\alpha$-approximate solution if it returns a set $S$ ($|S| \leq k$) such $f(S) \geq \alpha f(S^*)$. In the decision problem, the goal is to determine the optimal value, given a value OPT, we say an algorithm is $\alpha$-approximate if it can distinguish between $f(S^*) \geq \text{OPT}$ vs. $f(S^*) \leq \alpha \, \text{OPT}$.

We have the following basic fact about submodular functions.

**Fact 2.1.** *We have the following basic facts about submodular functions: (1) A linear function is submodular; (2) If $f, g$ are submodular, then $f + g$ is submodular; and (3) If $f$ is monotone submodular, Let $c > 0$ be any constant, and let $g(S) = \min\{f(S), c\}$ for any $S \subseteq [n]$, then $g$ is also submodular.*

**Information theory** To establish communication lower bounds, we need a few basic facts of information theory.

**Fact 2.2** (Hellinger v.s. total variation). *For any two distributions $P, Q$, we have*

$$h^2(P, Q) \leq \mathsf{TV}(P, Q) \leq \sqrt{2} h(P, Q).$$

**Fact 2.3.** *Suppose $A, B$ are independent when conditioned on $C$, then we have*

$$I(D; A|C) + I(D; B|C) \leq I(D; A, B|C)$$

## 3. Lower Bound for Submodular Maximization

We prove that the distributed set detection requires $\widetilde{\Omega}(n)$ communication cost in Section 3.1, using the distributed SDPI inequality and a direct sum argument. In Section 3.2, we give a simple reduction from distributed set detection to submodular maximization, which rules out the possibility of approximately *finding the optimal $k$-subset* using sublinear query (and it holds even for additive function). In Section 3.3, we present a more involved reduction that proves the hardness of *approximating the value of optimal $k$-subset*.

### 3.1. Communication Lower Bound

A key ingredient of our proof is a communication lower bound on the *distributed set detection* problem. The distributed set detection is a multi-party communication problem and we study it under the blackboard model, where every party can write and read over a common blackboard.

**Definition 3.1** (Distributed set detection). Let $n, k, m$ be input parameters, $\mathcal{D}_0, \mathcal{D}_1$ be two Bernoulli distributions with mean $\mu_0, \mu_1$. $m$ is the number of parties, who communicate in the blackboard model. Let $\mathcal{I} \subseteq [n]$ be an index set of size $[k/2, k]$. The input of the $t$-th party ($t \in [m]$) is a vector $X_t \in \{0, 1\}^n$, such that for any $i \in [n]$, $X_{t,i} \sim \mathcal{D}_0$ if $i \in [n] \setminus I$ and $X_{t,i} \sim \mathcal{D}_1$ if $i \in \mathcal{I}$. The goal is to output a set $\widehat{\mathcal{I}} \subseteq [n]$ ($|\widehat{\mathcal{I}}| = k$) that maximizes the intersection $|\mathcal{I} \cap \widehat{\mathcal{I}}|$.

The main goal of this section is to establish the following communication lower bound of distributed set detection.

**Theorem 3.2** (Communication lower bound). *Let $\epsilon \in (0, 1)$, $n, k, m$ be integers and satisfy $k \leq \epsilon n/4$. Let $c \geq 1$ be some constant and $\frac{1}{c}\mu_0 \leq \mu_1 \leq c\mu_0$. For the problem of distributed set detection, any (randomized) communication protocol that outputs $\epsilon$-fraction of the index set (i.e., $|\mathcal{I} \cap \widehat{\mathcal{I}}| \geq \epsilon k$) in expectation has communication complexity at least $\Omega(\epsilon^2 n/c)$.*

With the goal of proving Theorem 3.2, we first introduce the distributed detection problem, where each party only receives one coordinate and the goal is to detect whether they come from $\mathcal{D}_0$ or $\mathcal{D}_1$.

**Definition 3.3** (Distributed detection). Let $V \sim B_{1/2}$ and $\mathcal{D}_0, \mathcal{D}_1$ be two Bernoulli distributions with mean $\mu_0, \mu_1$. There are $m$ parties communicate in the blackboard model and each party receives a single bit $Z_t \sim \mathcal{D}_V$ independently drawn from $\mathcal{D}_V$ (with the same $V$ for all parties). The goal is to determine the value of $V$.

The distributed detection problem has large information cost.

- Using public randomness, the $m$ parties sample a set $\mathcal{I} = \{i_1, \ldots, i_k\} \subseteq [n]$.

- For any $t \in [m]$, the $t$-th party constructs the vector $X_t \in \{0,1\}^n$ as follow:

$$X_{t,i} \sim \begin{cases} \mathcal{D}_0 & i \in [n] \backslash \mathcal{I} \\ \mathcal{D}_1 & i \in \{i_2, \ldots, i_K\} \\ z_t & i = i_1 \end{cases}$$

- Then the $m$ parties follow the communication protocol of distributed set detection, and they output 1 if $i_1 \in \widehat{\mathcal{I}}$.

*Figure 1.* Communication protocol for distributed detection

**Lemma 3.4** (Distributed SDPI, Theorem 1.1 in (Braverman et al., 2016))**.** *Suppose $\frac{1}{c}\mu_0 \leq \mu_1 \leq c\mu_0$, $\beta(\mu_0, \mu_1) \leq 1$ be the SDPI constant of $\mu_0, \mu_1$. Let $\Pi$ be the communication transcript, $\Pi|_{V=0}$ (resp. $\Pi|_{V=1}$) be the transcript when $V = 0$ (resp. $V = 1$). In the distributed detection problem, we have the following distributed strong data processing inequality,*

$$h^2(\Pi|_{V=0}, \Pi|_{V=1}) \leq K \cdot c\beta(\mu_0, \mu_1) \cdot$$
$$\min\{I(\Pi; Z_1 \ldots Z_m|V=0), I(\Pi; Z_1 \ldots Z_m|V=1)\}$$

*for some fixed constant $K > 0$.*

We apply a direct sum argument and prove the information cost for distributed set detection is $\Omega(n)$ times the information cost of distributed detection. To this end, consider the communication protocol in Figure 1 for distributed detection.

Let $\Pi$ be the transcript of distributed set detection. Let $R$ be the public randomness used by the distributed set detection, $R' = (R, \mathcal{I})$ be the public randomness of distributed detection. The *information cost* of distributed set detection is defined as

$$\mathsf{IC} = \max_{\mathcal{I} \subseteq [n], k/2 \leq |\mathcal{I}| \leq k} I(\Pi; X_1, \ldots, X_m|\mathcal{I}, R). \quad (1)$$

The information cost is also a lower bound on the communication cost of distributed set detection.

We have the following direct-sum theorem on the information cost, the proof is similar to Proposition 5.2 in (Braverman et al., 2016).

**Lemma 3.5.** *For any $k \geq 2$, we have*

$$I(\Pi, R'; Z_1, \ldots, Z_m|V=0) \leq \frac{\mathsf{IC}}{n-k+1}.$$

We next analyse the correctness of the protocol.

**Lemma 3.6.** *Suppose the communication protocol of distributed set detection outputs $\epsilon$-fraction of the index set, then the protocol in Figure 1 correctly guesses the value of $V$ with probability at least $\frac{1}{2} + \frac{\epsilon}{4}$.*

Combining Lemma 3.4 – 3.6, we can prove Theorem 3.2.

### 3.2. Lower Bound for Search Problem

We start with a simpler linear query lower bound for the search problem. It is worth noting that the lower bound holds even when the function is additive.

**Theorem 3.7.** *Let $n$ be the size of the ground set, $k \in [n]$ be the cardinality constraint, $\alpha \in (0,1)$ be the approximation ratio and $k \leq O(\alpha n)$. A randomized algorithm must make at least $\Omega(\alpha^5 n/\log^2(n))$ queries in order to find an $\alpha$-approximate solution for submodular maximization under the cardinality constraint.*

We reduce from the distributed set detection problem. Let

$$\epsilon = \alpha/15, \qquad m = \frac{100 \log n}{\epsilon^2},$$

and

$$\mu_0 = \epsilon, \qquad \mu_1 = 1/2 \qquad c = 1/2\epsilon$$

Given an instance of distributed set detection with input $X_1, \ldots, X_m \in \{0,1\}^n$, consider the following function

$$f(S) = \sum_{i \in S} \sum_{t \in [m]} X_{t,i} \quad \forall S \subseteq \{0,1\}^n.$$

It is clear that the function $f$ is non-negative, monotone, submodular since it is a linear function. Furthermore, its optimal solution satisfies

**Lemma 3.8.** *With probability at least $1 - 1/n^{10}$, for any $\alpha$-approximate solution set $S \subseteq [n]$ ($|S| \leq k$), we have $|S \cap \mathcal{I}| \geq \epsilon k$.*

The proof of Theorem 3.7 follows from Lemma 3.8 and the communication lower bound of distribution set detection (Theorem 3.2).

### 3.3. Lower Bound for Decision Problem

We next prove a linear query lower bound for the decision problem.

**Theorem 3.9.** *Let $n$ be the size of the ground set, $k \in [n]$ be the cardinality constraint, $\alpha \in (0,1)$ be the approximation ratio, $\mathrm{OPT} \in \mathbb{R}^+$ and $k \leq O(\alpha^2 n)$. A randomized algorithm must make at least $\Omega(\alpha^{11} n/\log^2(n))$ queries in order to distinguish between*

- **YES Instance** $f(S^*) \geq \mathrm{OPT}$

- **NO Instance** $f(S^*) \leq \alpha\,\mathrm{OPT}$

We present a reduction from the distributed set detection problem. The construction of the submodular function and the choice of parameters are slightly different from Theorem 3.7. Let

$$\epsilon = \frac{\alpha^2}{800}, \qquad m = \frac{100 \log n}{\epsilon^2}$$

and

$$\mu_0 = \epsilon, \qquad \mu_1 = 1/2, \qquad c = 1/2\epsilon$$

Given an instance of distributed set detection with input $X_1, \ldots, X_m \in \{0,1\}^n$, consider the following functions

$$f_{\text{yes}}(S) = \min \Big\{ \sum_{i \in S \cap \mathcal{I}} \sum_{t \in [m]} X_{t,i} + \sum_{i \in S \cap [n] \setminus \mathcal{I}} \sum_{t \in [m]} X_{t,i} + \frac{\alpha m |S|}{20}, mk \Big\} \tag{2}$$

and

$$f_{\text{no}}(S) = \min \Big\{ \min \Big\{ \sum_{i \in S \cap \mathcal{I}} \sum_{t \in [m]} X_{t,i}, \frac{\alpha mk}{10} \Big\} + \sum_{i \in S \cap [n] \setminus \mathcal{I}} \sum_{t \in [m]} X_{t,i} + \frac{\alpha m |S|}{20}, mk \Big\}. \tag{3}$$

It is easy to see that both $f_{\text{yes}}, f_{\text{no}}$ are monotone and submodular, because linear combination and minimum with a constant keep submodularity and monotonicity (see Fact 2.1).

We first make a few simple observations.

**Lemma 3.10.** *With probability at least* $1 - 1/n^{10}$*, we have*

$$\sum_{t \in [m]} X_{t,i} \in [(1/2 - \epsilon)m, (1/2 + \epsilon)m] \quad \forall i \in \mathcal{I} \tag{4}$$

*and*

$$\sum_{t \in [m]} X_{t,i} \in (\epsilon m/2, 2\epsilon m) \quad \forall i \in [n] \setminus \mathcal{I} \tag{5}$$

In the rest of the proof, we would condition on the high probability event of Lemma 3.10. Let OPT $= mk/5$. We have the following observation on the optimal value of $f_{\text{yes}}$ and $f_{\text{no}}$.

**Lemma 3.11.** *We have (1)* $\max_{S^* \subseteq [n], |S^*| = k} f_{\text{yes}}(S^*) \geq$ OPT*; and* $f_{\text{no}}(S) \leq \alpha$ OPT *for any* $S \subseteq [n], |S| = k$*.*

Now we can proceed to prove Theorem 3.9.

*Proof of Theorem 3.9.* Suppose there exists an algorithm ALG that makes at most $R$ queries and distinguishes between the YES/NO instance. Consider the communication protocol in Algorithm 1 for distributed set detection.

From a high level, the communication proceeds in (at most) $R$ rounds. At the $r$-th round, the $m$ parties look at the $r$-th query $S_r \subseteq [n]$ of ALG, they either decide an output set or construct the value oracle $f(S_r)$. Concretely, if the size of $S_r$ is large, they set $f(S_r) = mk$ and proceed to the next round/query. Otherwise, they partition the set $S_r = S_{r,1} \cup \cdots \cup S_{r,20/\alpha}$ into $20/\alpha$ subsets, each of size at most $k$. The $m$ parties then compute the value of $\sum_{i \in S_{r,\tau}} \sum_{t \in [m]} X_{t,i}$ for each subset $S_{r,\tau}$ ($\tau \in [20/\alpha]$). If the value is large, they return the set $\widehat{I} \leftarrow S_{r,\tau}$; otherwise, they set the oracle value $f(S_r) = \min\{\sum_{t \in [m]} \sum_{i \in S_r} X_{t,i} + \alpha k |S_r|/20, mk\}$ and proceed to the next round/query.

---

**Algorithm 1** Reduction: From submodular maximization to distributed set detection

1: **Input:** Input $X_1, \ldots, X_m \in \{0,1\}^n$, submodular maximization algorithm ALG,
2: **for** $r = 1, 2, \ldots, R$ **do**
3:     Let $S_r \subseteq [n]$ be the $r$-th query of ALG
4:     **if** $|S_r| \geq 20k/\alpha$ **then**
5:         $f(S_r) \leftarrow mk$
6:     **else**
7:         Partition $S_r = S_{r,1} \cup \cdots \cup S_{r,20/\alpha}$
8:         **for** $\tau = 1, 2 \ldots, 20/\alpha$ **do**
9:             **if** $\sum_{i \in S_{r,\tau}} \sum_{t \in [m]} X_{t,i} \geq \alpha^2 mk/200$ **then**
10:             $\widehat{I} \leftarrow S_{r,\tau}$ and **return**
11:             **end if**
12:         **end for**
13:         $f(S_r) \leftarrow \min\{\sum_{i \in S_r} \sum_{t \in [m]} X_{t,i} + \alpha m |S_r|/20, mk\}$
14:     **end if**
15: **end for**

---

**Correctness** First, we prove the correctness of the protocol. We divide into two cases.

**Case 1.** Algorithm 1 returns a set $\widehat{\mathcal{I}} \leftarrow S_{r,\tau}$ at some round $r \in [R]$ and $\tau \in [20/\alpha]$. Then we prove it must satisfy $|S_{r,\tau} \cap \mathcal{I}| \geq \epsilon k$.

We have

$$\alpha^2 mk/200 \leq \sum_{i \in S_{r,\tau}} \sum_{t \in [m]} X_{t,i}$$
$$= \sum_{i \in S_{r,\tau} \cap \mathcal{I}} \sum_{t \in [m]} X_{t,i} + \sum_{i \in S_{r,\tau} \cap [n] \setminus \mathcal{I}} \sum_{t \in [m]} X_{t,i}$$
$$\leq |S_{r,\tau} \cap \mathcal{I}| \cdot (1/2 + \epsilon)m + k \cdot 2\epsilon m.$$

Here the first step follows from the assumption, the third step follows from Lemma 3.10. Plugging in the value of $\alpha, \epsilon$, we have that

$$|S_{r,\tau} \cap \mathcal{I}| \geq \frac{\alpha^2/200 - 2\epsilon}{1/2 + \epsilon} k \geq \epsilon k.$$

**Case 2.** Suppose Algorithm 1 never returns anything, i.e., the if condition at Line 9 of Algorithm 1 is never satisfied. Then we prove $f(S_r) = f_{\text{yes}}(S_r) = f_{\text{no}}(S_r)$ holds for every $r \in [R]$.

Fix a round $r \in [R]$, if $|S_r| \geq 20k/\alpha$, then we have $f_{\text{yes}}(S_r) = f_{\text{no}}(S_r) = mk = f(S_r)$ due to Line 5 of Algorithm 1. Now suppose $|S_r| < \frac{20k}{\alpha}$, we have

$$f(S_r) = \min\left\{\sum_{i \in S_r}\sum_{t \in [m]} X_{t,i} + \alpha m|S_r|/20, mk\right\}$$
$$= f_{\text{yes}}(S_r)$$

Meanwhile, by the assumption of the second case, we also have

$$\sum_{i \in S_r \cap \mathcal{I}}\sum_{t \in [m]} X_{t,i} \leq \sum_{i \in S_r}\sum_{t \in [m]} X_{t,i}$$
$$= \sum_{\tau \in [20/\alpha]}\sum_{i \in S_{r,\tau}}\sum_{t \in [m]} X_{t,i}$$
$$\leq (20/\alpha) \cdot (\alpha^2 mk/200) = \alpha mk/10,$$

Hence, we also have $f_{\text{no}}(S_r) = f_{\text{yes}}(S_r)$ (see the definition in Eq. (2)(3)). This proved that $f_{\text{yes}}(S_r) = f_{\text{no}}(S_r) = f(S_r)$ for any $r \in [R]$.

In summary, we conclude that in Case 1, the reduction outputs a set $\widehat{\mathcal{I}}$ with $|\widehat{\mathcal{I}} \cap \mathcal{I}| \geq \epsilon k$, while in Case 2, the transcript of ALG is the same for $f_{\text{yes}}$ and $f_{no}$. Since we assume ALG is able to distinguish between $f_{\text{yes}}, f_{\text{no}}$ after $R$ queries, then we must fall into Case 1. This proves the correctness of Algorithm 1.

**Communication complexity** Next we analyse the communication complexity. At each round $r \in [R]$, the $m$ parties need to compute $\sum_{t \in [m]}\sum_{i \in S_{r,\tau}} X_{t,i}$ for $\tau \in [20/\alpha]$. For each $\tau \in [20/\alpha]$, the $t$-th party ($t \in [m]$) could compute the partial sum $\sum_{i \in S_{r,\tau}} X_{t,i}$ locally, and then write it on the blackboard. Hence, the total communication cost per round is at most $\tau \cdot m \cdot \log(n) = O(\log^2(n)/\alpha\epsilon^2) = O(\log^2(n)/\alpha^5)$ There are at most $R$ rounds, so the total communication cost over $R$ rounds are $O(R\log^2(n)/\alpha^5)$. By Theorem 3.2, we must have

$$R\log^2(n)/\alpha^5 \geq \Omega(\epsilon^2 n/c) \quad \Rightarrow \quad R \geq \Omega(\alpha^{11}n/\log^2(n)).$$
$\square$

## 4. Sublinear Algorithm for Additive Function

**Theorem 4.1.** *Let $n$ be the size of ground set, $k$ be the constraint, $\epsilon \in (0, 1/8)$ be a constant. Suppose $f$ is monotone and additive, then there is an algorithm that approximates the value of $\max_{S^* \in [n], |S| = k} f(S)$ within $(1 \pm \epsilon)$ factor using $(n/k) \cdot \text{poly}(\log(n), \epsilon^{-1})$ number of queries and succeeds with probability at least $4/5$.*

**Parameters and notation** Let $w_i = f(i) \geq 0$ and we have $f(S) = \sum_{i \in S} w_i$ for all $S \subseteq [n]$. For parameters $R, R_1, R_2$ defined below, we define a set $\{k_r\}_r$ of scales ranging from $k_1 = 1$ to $k_{R+R_1+R_2} = k$. Specifically, we let $R = 100\log(n)/\epsilon^2$, $R_1 = \log_{1+\epsilon}(k/R^3)$, $R_2 = \log_{1+\epsilon}(R^2)$. Let $k_r = r$ for $r \leq R$, and $k_r = R(1+\epsilon)^{r-R}$ for $r \in [R+1 : R+R_1+R_2]$.

**Algorithm description** Our approach is depicted as LINEARSUM (Algorithm 2). From a high level, LINEARSUM divides the largest $k$ elements into multiple scales. For the largest scales, $k_r \in \{k_{R+R_1+1}, \ldots, k_{R+R_1+R_2}\}$, LINEARSUM directly estimates the $k_r$-th quantile and it takes roughly $\widetilde{O}(n/k_r) = \widetilde{O}(n/k)$ samples.

For smaller scales, $k_r \in \{k_1, \ldots, k_{R+R_1}\}$, we cannot directly use this naive estimation approach as we cannot afford $\widetilde{O}(n/k_r)$ query complexity for smaller $k_r$. To avoid this increase in query complexity, LINEARSUM randomly partitions the ground set $[n]$ into $nk_r/k$ buckets $A_{r,1}, \ldots, A_{r,nk_r/l}$. Define $f_r(i) := f(A_{r,i})$ for $i \in [nk_r/k]$, LINEARSUM estimates the $k_r$-th largest element of $f_r$. Roughly speaking, we expect the top $k_r$-th bucket to be as valuable as the top $k_r$-th element + an average bucket, so LINEARSUM further subtracts the average value of a bucket to get an estimate of the contribution just from the top $k_r$ elements.

---

**Algorithm 2** LINEARSUM$(f, k)$

1: **for** $r = 1, 2, \ldots, R + R_1$ **do**
2:     Random partition $[n] = A_{r,1} \cup \cdots \cup A_{r,nk_r/k}$ into $nk_r/k$ subsets
3:     Define $f_r(i) := f(A_{r,i})$ for $\forall i \in [nk_r/k]$
4:     $b_r \leftarrow$ ESTIMATEQUANTILE$(f_r, k_r)$
5:     $c_r \leftarrow b_r - \frac{k}{nk_r}f([n])$
6: **end for**
7: $c_r \leftarrow$ ESTIMATEQUANTILE$(f, k_r)$ for $r \in [R + R_1 + 1 : R + R_1 + R_2]$
8: **Return** $\sum_{r=1}^{R+R_1+R_2}(k_r - k_{r-1})c_r$

---

**Algorithm 3** ESTIMATEQUANTILE$(g, t)$

1: **if** $t > 100\log(n)/\epsilon^2$ **then**
2:     Random sample a set $S \subseteq [m]$ of size $100m\log(n)/t\epsilon^2$
3:     Query $g(i)$ for all $i \in S$ and let $b_t$ be the $100\log(n)/\epsilon^2$-th largest element of $\{g(i)\}_{i \in S}$
4:     **Return** $b_t$
5: **else**
6:     Query $g$ and **Return** the $t$-th largest element
7: **end if**

## 4.1. Analysis

We relabel the ground set elements such that $w_1 \geq \cdots \geq w_n$ for convenience; note that LINEARSUM is oblivious to the labeling of ground set elements so this is without loss of generality.

We first prove that ESTIMATEQUANTILE$(g, t)$ gives a good estimate on the value of the top $t$-th element of $g$.

**Lemma 4.2.** *Given any function* $g : [m] \to \mathbb{R}^+$, *and let* $\widehat{b}_1 \geq \widehat{b}_2 \geq \cdots \widehat{b}_m$ *be the descending ordering of* $\{g(i)\}_{i \in [m]}$. *For any* $t \in [m]$, *the output* $b_t$ *of* ESTIMATEQUANTILE$(g, t)$ *satisfies*

$$\widehat{b}_{(1+\epsilon)t} \leq b_t \leq \widehat{b}_{(1+\epsilon)^{-1}t}$$

*for* $t > 100 \log(n)/\epsilon^2$ *with probability at least* $1 - 1/n^4$, *and* $b_t = \widehat{b}_t$ *for* $t \leq 100 \log(n)/\epsilon^2$.

In the rest of the proof, it is convenient to assume $k$ divides $n$, and further that

$$(\log(n)/\epsilon)^8 \leq k \leq n \cdot (\epsilon/\log(n))^8. \tag{6}$$

This assumption is without loss of generality as we could just add dummy elements to the ground set.

The key step is to prove that $c_r$ gives a good estimate on the $k_r$-th largest element of $f$.

**Lemma 4.3.** *For* $r \leq R$, *with probability at least* $1 - \frac{1}{100R}$

$$w_{k_r} - \frac{2\epsilon}{R} \sum_{j \leq k} w_j \leq c_r \leq w_{k_r} + \frac{2\epsilon}{R} \sum_{j \leq k} w_j. \tag{7}$$

*and for* $r \in [R + 1 : R + R_1]$, *with probability at least* $1 - \frac{1}{n^4}$, *we have*

$$w_{(1+2\epsilon)k_r} - \frac{2}{k_r} \sum_{j \leq k} w_j \leq c_r \leq w_{(1-2\epsilon)k_r} + \frac{2}{k_r} \sum_{j \leq k} w_j. \tag{8}$$

The detailed proof of Lemma 4.3 can be found at Appendix B and we sketch the high level idea here. In the proof of Lemma 4.3, fix a value of $r$, let $t_r = \log^3(n)/\epsilon^3$ when $r \leq R$ and $t_r = k_r \log^2(n)/\epsilon$ when $r \in [R + 1 : R + R_1]$. For each subset $i \in [nk_r/k]$, we decompose $f(A_{r,i})$ into three parts

$$f(A_{r,i}) = \sum_{j \in A_{r,i}} w_j$$

$$= \sum_{j \in A_{r,i}} w_j \mathbf{1}_{j \leq t_r} + \sum_{j \in A_{r,i}} w_j \mathbf{1}_{t_r < j \leq k} + \sum_{j \in A_{r,i}} w_j \mathbf{1}_{j > k}.$$

From a high level, we wish to prove that (1) there is at most one element contributes the first term; (2) the second term is negligible; and (3) the last term concentrates on its mean.

Fix a partition $A_{r,1} \cup \cdots \cup A_{r,nk_r/k}$. For any subset $S \subseteq [n]$, the number of collisions among $S$ is defined as the total number of elements in $S$ that are allocated to subsets with more than one element of $S$. We first prove that (with sufficiently high probability) there are few collisions among the largest $t_r$ elements.

**Lemma 4.4.** *We have*

- *For* $r \leq R$, *with probability at least* $1 - \frac{1}{100R}$, *there are no collisions among the top* $t_r$ *elements.*

- *For* $r \in [R + 1 : R + R_1]$, *with probability at least* $1 - 1/n^4$, *the number of collisions among the top* $t_r$ *elements are at most is at most* $\epsilon k_r$

We next prove $\sum_{j \in A_{r,i}} w_j \mathbf{1}_{t_r < j \leq k}$ is negligible.

**Lemma 4.5.** *For any* $r \in [R + R_1]$, $i \subseteq [nk_r/k]$, *with probability at least* $1 - 1/n^4$, *we have*

$$\sum_{j \in A_{r,i}} w_j \mathbf{1}_{t_r < j \leq k} \leq \frac{\log^2(n)}{t_r} \sum_{j \leq k} w_j.$$

Finally, we prove $\sum_{j \in A_{r,i}} w_j \cdot \mathbf{1}_{j > k}$ concentrates around the mean.

**Lemma 4.6.** *For any* $r \in [R + R_1]$ *and* $i \in [nk_r/k]$, *with probability at least* $1 - 1/n^5$,

$$\sum_{j \in A_{i,r}} w_j \cdot \mathbf{1}_{j > k} = \frac{k}{nk_r} \sum_{j > k} w_j \pm \sqrt{\frac{k}{k_r}} \log(n) w_k.$$

We can obtain Lemma 4.3 from Lemma 4.4 – Lemma 4.6, and obtain Theorem 4.1 using Lemma 4.3.

### 4.2. A Nearly-matching Lower Bound

We present a matching lower bound to Theorem 4.1 whose proof can be found at the Appendix.

**Theorem 4.7.** *Let* $f$ *be an additive function,* $n$ *be the size of ground set,* $k$ *be the constraint,* $\alpha \in (0,1)$ *be a constant. Then it takes* $\Omega(\alpha^3 n/k \log(n))$ *queries to distinguish between*

- **YES Instance:** $f(S^*) \geq \text{OPT}$

- **NO Instance:** $f(S^*) \leq \alpha \text{OPT}$

### Acknowledgement

The authors would like to thank Yair Carmon for useful discussion over the project. The work is supported by NSF CCF-1954927, a David and Lucile Packard Fellowship, AFOSR award FA95502310251 and ONR award number N000142212771.

## Impact Statement

This is a theoretical paper and it does not present foreseeable negative social impacts.

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

# A. Missing Proof from Section 3

We first prove Lemma 3.5.

*Proof of Lemma 3.5.* First, we have

$$
\begin{aligned}
I(\Pi, R'; Z_1, \ldots, Z_m | V = 0) &= I(\Pi; Z_1, \ldots, Z_m | R', V = 0) \\
&= I(\Pi; X_{1,\mathcal{I}_1}, \ldots, X_{m,\mathcal{I}_1} | R, \mathcal{I}, V = 0) \\
&= \mathop{\mathbb{E}}_{i_2, \ldots, i_k} [I(\Pi; X_{1,\mathcal{I}_1}, \ldots, X_{m,\mathcal{I}_1} | R, \mathcal{I}_1, \mathcal{I}_2 = i_2, \ldots, \mathcal{I}_k = i_k, V = 0). \quad (9)
\end{aligned}
$$

The first step follows because the public randomness $R'$ is independent of $Z_1, \ldots, Z_m$ conditioning on $V = 0$, the second step follows from $R' = (R, \mathcal{I})$, and $Z_1, \ldots, Z_m$ are embedded to the $\mathcal{I}_1$-th coordinate. The third step follows from the definition of condition mutual information.

For any fixed $i_2, \ldots, i_k$, we bound the RHS of Eq. (9) and our goal is to prove

$$
I(\Pi; X_{1,\mathcal{I}_1}, \ldots, X_{m,\mathcal{I}_1} | R, \mathcal{I}_1, \mathcal{I}_2 = i_2, \ldots, \mathcal{I}_k = i_k, V = 0) \le \frac{\mathsf{IC}}{n - k + 1}. \quad (10)
$$

To this end, we have

$$
\begin{aligned}
&I(\Pi; X_{1,\mathcal{I}_1}, \ldots, X_{m,\mathcal{I}_1} | R, \mathcal{I}_1, \mathcal{I}_2 = i_2, \ldots, \mathcal{I}_k = i_k, V = 0) \\
&= \frac{1}{n - k + 1} \sum_{i \in [n] \setminus \{i_2, \ldots, i_k\}} I(\Pi; X_{1,\mathcal{I}_1}, \ldots, X_{m,\mathcal{I}_1} | R, \mathcal{I}_1 = i, \mathcal{I}_2 = i_2, \ldots, \mathcal{I}_k = i_k, V = 0) \\
&= \frac{1}{n - k + 1} \sum_{i \in [n] \setminus \{i_2, \ldots, i_k\}} I(\Pi; X_{1,i}, \ldots, X_{m,i} | R, \mathcal{I}_2 = i_2, \ldots, \mathcal{I}_k = i_k, V = 0) \\
&\le \frac{1}{n - k + 1} I(\Pi; \{X_{1,i}, \ldots, X_{m,i}\}_{i \in [n] \setminus \{i_2, \ldots, i_k\}} | R, \mathcal{I}_2 = i_2, \ldots, \mathcal{I}_k = i_k, V = 0) \\
&\le \frac{1}{n - k + 1} I(\Pi; X_1, \ldots, X_m | R, \mathcal{I}_2 = i_2, \ldots, \mathcal{I}_k = i_k, V = 0). \quad (11)
\end{aligned}
$$

The first step holds since the choice of $I_1$ is uniform over $[n] \setminus \{i_2, \ldots, i_k\}$. The second step holds since the distribution of $X_{1,i}, \ldots, X_{m,i}$ for $i \in [n] \setminus \{i_2, \ldots, i_k\}$ does not depend on $i$ (because they are drawn from $\mathcal{D}_0^m$), and the transcript $\Pi$ is oblivious of $\mathcal{I}_1$. The third step holds due to $X_{1,i}, \ldots X_{m,i}$ are independent across $i \in [n] \setminus \{i_2, \ldots, i_k\}$ and Fact 2.3.

Finally, for any $i_2, \ldots, i_k$, we have

$$
I(\Pi; X_1, \ldots, X_m | R, \mathcal{I}_2 = i_2, \ldots, \mathcal{I}_k = i_k, V = 0) \le \mathsf{IC} \quad (12)
$$

holds for any $k \ge 2$ due to the definition of information cost (see Eq. (1)).

We have proved Eq. (10) combining Eq. (11)(12), and combining with Eq. (9), we complete the proof. $\square$

We next prove Lemma 3.6

*Proof of Lemma 3.6.* When $V = 1$, note $\{X_{1,i}, \ldots, X_{m,i}\}_{i \in \mathcal{I}}$ are drawn from the same distribution, hence we have

$$
\Pr[i_1 \in \mathcal{I}] = \frac{|\mathcal{I} \cap \widehat{\mathcal{I}}|}{|\mathcal{I}|} \ge \frac{\epsilon k}{|\mathcal{I}|} \ge \epsilon.
$$

When $V = 0$, note $\{X_{1,i}, \ldots, X_{m,i}\}_{i \in [n] \setminus \{i_2, \ldots, i_k\}}$ have the same distribution, hence we have

$$
\Pr[i_1 \in \mathcal{I}] = \frac{|\widehat{\mathcal{I}}|}{n - k + 1} = \frac{k}{n - k + 1}
$$

Combining the above two cases, the success probability is at least

$$
\frac{1}{2} \cdot \epsilon + \frac{1}{2} \cdot (1 - \frac{k}{n - k + 1}) \ge \frac{1}{2} + \frac{\epsilon}{4}.
$$

Here we use the fact that $k \le \epsilon n / 4$. $\square$

Now we provide the proof of Theorem 3.2

*Proof of Theorem 3.2.* By Lemma 3.6, we have that $\mathsf{TV}(\Pi|_{V=0}, \Pi|_{V=1}) \geq \frac{\epsilon}{4}$, and therefore,

$$h^2(\Pi|_{V=0}, \Pi|_{V=1}) \geq \frac{1}{2}\mathsf{TV}^2(\Pi|_{V=0}, \Pi|_{V=1}) \geq \frac{\epsilon^2}{32}.$$

Here, the first step follows from Fact 2.2.

By Lemma 3.4 and the fact that the SDPI constant $\beta(\mu_0, \mu_1)$ is at most 1, we have

$$I(\Pi; Z_1, \ldots, Z_m | V = 0) \geq \Omega(1/c) \cdot h^2(\Pi|_{V=0}, \Pi|_{V=1}) = \Omega(\epsilon^2/c).$$

By Lemma 3.5, we have $\mathsf{IC} \geq \Omega(\epsilon^2 n/c)$. This completes the proof since the communication cost is at least the information cost. $\qquad\square$

Next we prove lower bound for submodular maximization. We first prove Lemma 3.8.

*Proof of Lemma 3.8.* By Chernoff bound, we have

$$f(i) = \sum_{t \in [m]} X_{t,i} \in [(1/2 - \epsilon)m, (1/2 + \epsilon)m] \quad \forall i \in \mathcal{I} \tag{13}$$

and

$$f(i) = \sum_{t \in [m]} X_{t,i} \leq 2\epsilon m \quad \forall i \in [n] \setminus \mathcal{I} \tag{14}$$

holds with probability at least $1 - 1/n^{10}$. Hence, we have

$$f(S^*) \geq f(\mathcal{I}) = \sum_{i \in \mathcal{I}} f(i) \geq (k/2)(1/2 - \epsilon)m. \tag{15}$$

The first two steps follow from the definition, the last step follows from $|\mathcal{I}| \geq k/2$ and Eq. (13).

On the other hand, for any set $S \subseteq [n]$ with size $k$, we have

$$f(S) = f(S \cap \mathcal{I}) + f(S \setminus \mathcal{I}) \leq |S \cap \mathcal{I}| \cdot (\frac{1}{2} + \epsilon) \cdot m + k \cdot 2\epsilon m. \tag{16}$$

The first step follows from the definition of $f$. The second step follows from Eq. (13)(14). Combining Eq. (15)(16), we can conclude that for any $\alpha$-approximate solution $S$,

$$\frac{f(S)}{f(S^*)} \geq \alpha = 15\epsilon \quad \Rightarrow \quad \frac{|S \cap \mathcal{I}| \cdot (\frac{1}{2} + \epsilon) \cdot m + k \cdot 2\epsilon m}{(k/2)(1/2 - \epsilon)m} \geq 15\epsilon \quad \Rightarrow \quad |S \cap \mathcal{I}| \geq \epsilon k.$$

$\qquad\square$

We next prove Theorem 3.7

*Proof of Theorem 3.7.* Suppose there exists an algorithm $\mathrm{ALG}$ that makes at most $R$ queries and outputs an $\alpha$-approximate solution $S$ for the submodular maximization problem. Consider the following communication protocol: The protocol proceeds in $R$ rounds, where in the $r$-th round ($r \in [R]$), suppose $\mathrm{ALG}$ queries set $S_r$, then the $m$ parties collectively compute the value of $f(S_r)$. Since

$$f(S_r) = \sum_{i \in S_r} \sum_{t \in [m]} X_{t,i} = \sum_{t \in [m]} \sum_{i \in S_r} X_{t,i},$$

it suffices for the $t$-th party to compute $\sum_{i \in S_r} X_{t,i}$ locally and writes it on the blackboard. Given the knowledge of $S_1, \ldots, S_r, f(S_1), \ldots, f(S_r)$, $m$ parties can simulate ALG and determine the next query $S_{r+1}$, and therefore, continue the protocol. Finally, $m$ parties output the solution set $\widehat{\mathcal{I}} = S$.

The communication cost at each round equals $m \cdot \log(n) = O(\log^2(n)/\epsilon^2)$, and there is a sequence of $R$ queries, so there are $O(R \log^2(n)/\epsilon^2)$ bits of communication in total. Moreover, ALG guarantees the output solution $\widehat{\mathcal{I}} = S$ is $\alpha = 15\epsilon$-approximate, by Lemma 3.8, we know that $|\widehat{\mathcal{I}} \cap \mathcal{I}| \geq \epsilon k$. By Theorem 3.2, we must have

$$R \log^2(n)/\epsilon^2 \geq \Omega(\epsilon^2 n/c) \quad \Rightarrow \quad R \geq \Omega(\epsilon^5 n/\log^2(n)) = \Omega(\alpha^5 n/\log^2(n)).$$

$\square$

We next prove Lemma 3.10 and Lemma 3.11.

*Proof of Lemma 3.10.* This follows directly from Chernoff bound. For any $i \in \mathcal{I}$, $X_{t,i} \sim B_{1/2}$, and therefore

$$\Pr\left[\left|\sum_{t \in [m]} X_{t,i} - m/2\right| \geq \epsilon m\right] \leq 2\exp(-2m\epsilon^2) = \frac{2}{n^{200}}$$

and for any $i \in [n] \backslash \mathcal{I}$, $X_{t,i} \sim B_\epsilon$,

$$\Pr\left[\left|\sum_{t \in [m]} X_{t,i} - \epsilon m\right| \geq \epsilon m/2\right] \leq 2\exp(-m\epsilon^2/2) = \frac{2}{n^{50}}.$$

$\square$

*Proof of Lemma 3.11.* For the YES instance, we have

$$\max_{S^* \subseteq [n], |S^*|=k} f_{\text{yes}}(S^*) \geq f_{\text{yes}}(\mathcal{I}) \geq \min\left\{\sum_{i \in \mathcal{I}} \sum_{t \in [m]} X_{t,i}, mk\right\} \geq (k/2)(1/2 - \epsilon)m \geq \text{OPT}$$

where the second step follows from the definition of $f_{\text{yes}}$ (see Eq. (2)), the third step follows from Eq. (4) and the last step holds since OPT.

For the NO instance, for any set $S$ of size at most $k$, we have

$$f_{\text{no}}(S) \leq \min\left\{\sum_{i \in S \cap \mathcal{I}} \sum_{t \in [m]} X_{t,i}, \frac{\alpha mk}{10}\right\} + \sum_{i \in S \cap [n] \backslash \mathcal{I}} \sum_{t \in [m]} X_{t,i} + \frac{\alpha mk}{20}$$
$$\leq \alpha mk/10 + k \cdot 2\epsilon m + \alpha mk/20 \leq \alpha \, \text{OPT}$$

where the first step follows from the definition of $f_{\text{no}}$ (see Eq. (3)), and the second step holds due to Eq. (5) and $|S| \leq k$. The last step follows from the choice of parameters. $\square$

# B. Missing Proof from Section 4

We first prove Lemma 4.2.

*Proof of Lemma 4.2.* We focus on the non-trivial case of $t > 100\log(n)/\epsilon^2$. By Chernoff bound, we have

$$\Pr[b_t \leq \widehat{b}_{(1+\epsilon)t}] = \Pr\left[|S \cap [(1+\epsilon)t]| \leq \frac{100\log(n)}{\epsilon^2}\right] = \exp\left(-\epsilon^2 \cdot \frac{100\log(n)}{\epsilon^2} \cdot \frac{1}{2}\right) \leq \frac{1}{n^{50}}$$

here the second step follows from Chernoff bound and

$$\mathbb{E}[|S \cap [(1+\epsilon)t]|] = \frac{100m \log(n)}{t\epsilon^2} \cdot \frac{(1+\epsilon)t}{m} = (1+\epsilon) \cdot \frac{100 \log(n)}{\epsilon^2}$$

Similarly,

$$\Pr[b_t \geq \widehat{b}_{(1+\epsilon)^{-1}t}] = \Pr\left[|S \cap [(1+\epsilon)^{-1}t]| \geq \frac{100 \log(n)}{\epsilon^2}\right]$$

$$= \exp\left(-\left(\frac{\epsilon}{1+\epsilon}\right)^2 \cdot \frac{100 \log(n)}{(1+\epsilon)\epsilon^2} \cdot \frac{1}{3}\right) \leq \frac{1}{n^4}.$$

where the second step follows from Chernoff bound and

$$\mathbb{E}[|S \cap [(1+\epsilon)^{-1}t]|] = \frac{100m \log(n)}{t\epsilon^2} \cdot \frac{t}{(1+\epsilon)m} = \frac{100 \log(n)}{(1+\epsilon)\epsilon^2}.$$

$\square$

We next prove Lemma 4.4 – 4.6

*Proof of Lemma 4.4.* For $r \leq R$, the probability that is there are no collisions among $[t_r]$ equals

$$1 \cdot \left(1 - \frac{k}{k_r n}\right) \cdots \left(1 - \frac{(t_r - 1)k}{k_r n}\right) \geq 1 - t_r^2 \cdot \frac{k}{k_r n} \geq 1 - \frac{1}{100R}$$

(here the last inequality uses (6)).

For $r \in [R+1 : R + R_1]$, consider the random process that $j = 1, 2, 3, \ldots, t_r$ are randomly put into $A_{r,1}, \ldots, A_{r,nk_r/k}$. Let $X_j$ indicates if $j$ falls into the same set as some element $j' < j$, i.e.,

$$X_j = \begin{cases} 1 & j \in A_{r,i}, j' \in A_{r,i} \text{ for some } j' < j, i \in [nk_r/k] \\ 0 & \text{otherwise} \end{cases}$$

We know that $\mathbb{E}[X_j] \leq t_r \cdot \frac{k}{k_r n}$, and

$$\mathbb{E}\left[\sum_{j \leq t_r} X_t\right] \leq t_r^2 \cdot \frac{k}{k_r n} = (k_r \log^2(n)/\epsilon)^2 \cdot \frac{k}{k_r n} = \frac{k \log^4(n)}{n\epsilon^2} \cdot k_r \leq \epsilon k_r / 4$$

where the last step holds since we assume $k \leq \epsilon^8 n / \log^8(n)$.

By Azuma-Hoeffding bound, we have

$$\Pr\left[\sum_{j \leq t_r} X_j \geq \epsilon k_r / 2\right] \leq \exp(-\epsilon k_r / 12) \leq 1/n^5.$$

This completes the proof. $\square$

*Proof of Lemma 4.5.* For any $r \in [R + R_1]$, for any subset $i \subseteq [nk_r/k]$, we have

$$\mathbb{E}\left[\sum_{j \in A_{i,r}} w_j \mathbb{1}_{t_r < j \leq k}\right] = \frac{k}{nk_r} \sum_{t_r < j \leq k} w_j \leq \frac{\log^2(n)}{2t_r} \sum_{j \leq k} w_j,$$

where the second step follows from the choice of parameters. By Chernoff bound, we have

$$\Pr\left[\sum_{j \in A_{r,i}} w_j \mathbb{1}_{t_r < j \leq k} \geq \frac{\log^2(n)}{t_r} \sum_{j \leq k} w_j\right] \leq \exp\left(-\sum_{j \leq k} w_j \log^2(n)/6t_r w_{t_r}\right)$$

$$\leq \exp(-\log^2(n)/6) \leq 1/n^6$$

where the second step follows from $w_{t_r} \leq \frac{1}{t_r} \sum_{j \leq t_r} w_j \leq \frac{1}{t_r} \sum_{j \leq t} w_j$. $\square$

*Proof Lemma 4.6.* We have

$$\mathbb{E}\left[\sum_{j \in A_{r,i}} w_j \cdot 1_{j>k}\right] = \frac{k}{nk_r} \sum_{j>k} w_j,$$

We use Chernoff bound. If $\frac{k}{nk_r} \sum_{j>k} w_j \geq \sqrt{k/k_r} \log(n) w_k$, then we have

$$\Pr\left[\left|\sum_{j \in A_{r,i}} w_j \cdot 1_{j>k} - \frac{k}{nk_r} \sum_{j>k} w_j\right| \leq \sqrt{k/k_r} \log(n) w_k\right]$$

$$\geq 1 - 2\exp\left(-\frac{\log^2(n)k/k_r}{3k \sum_{j>k} w_j/nk_r w_k}\right)$$

$$\geq 1 - \exp(-\log^2(n)/3) \geq 1 - \frac{1}{n^5}.$$

Otherwise, If $\frac{k}{nk_r} \sum_{j>k} w_j < \sqrt{k/k_r} \log(n) w_k$, then we have

$$\Pr\left[\left|\sum_{j \in A_{r,i}} w_j \cdot 1_{j>k} - \frac{k}{nk_r} \sum_{j>k} w_j\right| \leq \sqrt{k/k_r} \log(n) w_k\right]$$

$$\geq 1 - 2\exp\left(-\log^2(n)k/3k_r\right)$$

$$\geq 1 - \exp(-\log^2(n)/3) \geq 1 - \frac{1}{n^5}.$$

$\square$

Now we can finish the proof of Lemma 4.3.

*Proof of Lemma 4.3.* Fix a value of $r$, for each subset $i \in [nk_r/k]$, we would decompose $f(A_{r,i})$ into three parts

$$f(A_{r,i}) = \sum_{j \in A_{r,i}} w_j = \sum_{j \in A_{r,i}} w_j 1_{j \leq t_r} + \sum_{j \in A_{r,i}} w_j 1_{t_r < j \leq k} + \sum_{j \in A_{r,i}} w_j 1_{j>k}. \tag{17}$$

CASE 1: $r \in [R+1 : R+R_1]$

We first consider the case that $r \in [R+1 : R+R_1]$ and prove Eq. (8). For the LHS of Eq. (8), let $\mathcal{I} \subseteq [nk_r/k]$ be the collection of subsets such that contain the top $(1+2\epsilon)k_r$ elements, i.e. $[(1+2\epsilon)k_r] \subseteq \cup_{i \in \mathcal{I}} A_{r,i}$. By Lemma 4.4, we know that $|\mathcal{I}| \geq (1+\epsilon)k_r$. We have

$$b_r \geq \min_{i \in \mathcal{I}}\left\{\sum_{j \in A_{r,i}} w_j\right\}$$

$$= \min_{i \in \mathcal{I}}\left\{\sum_{j \in A_{r,i}} w_j 1_{j \leq t_r} + \sum_{j \in A_{r,i}} w_j 1_{t_r < j \leq k} + \sum_{j \in A_{r,i}} w_j 1_{j>k}\right\}$$

$$\geq w_{(1+2\epsilon)k_r} + 0 + \frac{k}{nk_r} \sum_{j>k} w_j - \sqrt{k/k_r} \log(n) w_k$$

$$\geq w_{(1+2\epsilon)k_r} + \frac{k}{nk_r} \sum_{j>k} w_j - \frac{1}{k_r} \sum_{j \leq k} w_j \tag{18}$$

The first step follows from the guarantee of QUANTILEESTIMATE (see Lemma 4.2), the third step follows from Lemma 4.6, the last step follows from $k_r \leq k/\log^2(n)$.

Meanwhile, we have

$$\frac{k}{nk_r}f([n]) = \frac{k}{nk_r}\sum_{j=1}^{n}w_j = \frac{k}{nk_r}\sum_{j\leq k}w_j + \frac{k}{nk_r}\sum_{j>k}w_j \leq \frac{1}{k_r}\sum_{j\leq k}w_j + \frac{k}{nk_r}\sum_{j\geq k}w_j \tag{19}$$

Combining Eq. (18)(19), we have

$$c_r = b_r - \frac{k}{nk_r}f([n]) \geq w_{(1+2\epsilon)k_r} - \frac{2}{k_r}\sum_{j\leq k}w_r$$

For the RHS of Eq. (8). Let $\mathcal{I}_r \subseteq [nk_r/k]$ be the top $(1-\epsilon)k_r$ subsets of $\{A_{r,i}\}_{i\in[nk_r/k]}$. By Lemma 4.4, we know that there exists $i_r \in \mathcal{I}_r$, such that, either $|A_{r,i_r} \cap [t_r]| = 0$, or $j_r = A_{r,i_r} \cap [t_r]$ and $j_r \geq (1-2\epsilon)k_r$. Therefore, we have

$$
\begin{aligned}
b_r &\leq \sum_{j\in A_{r,i_r}}w_j \\
&= \sum_{j\in A_{r,i_r}}w_j 1_{j\leq t_r} + \sum_{j\in A_{r,i_r}}w_j 1_{t_r<j\leq k} + \sum_{j\in A_{r,i_r}}w_j 1_{j>k} \\
&\leq w_{(1-2\epsilon)k_r} + \frac{\log^2(n)}{t_r}\sum_{j\leq k}w_j + \frac{k}{nk_r}\sum_{j>k}w_j + \sqrt{k/k_r}\log(n)w_k \\
&\leq w_{k_r} + \frac{k}{nk_r}\sum_{j>k}w_j + \frac{1+1}{k_r}\sum_{j\leq k}w_j
\end{aligned}
$$

The first step follows from the guarantee of QuantileEstimate (see Lemma 4.2), the third step follows from Lemma 4.5 and Lemma 4.6, the last step follows from the choice of parameters.

Hence, we have

$$
\begin{aligned}
c_r &= b_r - \frac{k}{nk_r}f([n]) \\
&\leq w_{(1-2\epsilon)k_r} + \frac{k}{nk_r}\sum_{j>k}w_j + \frac{2}{k_r}\sum_{j\leq k}w_j - \frac{k}{nk_r}\sum_{j\geq k}w_j \\
&\leq w_{(1-2\epsilon)k_r} + \frac{2}{k_r}\sum_{j\leq k}w_j.
\end{aligned}
$$

CASE 2: $r \leq R$

We next study the case that $r \leq R$. The proof is similar. For the LHS of Eq. (7), let $\mathcal{I} \subseteq [nk_r/k]$ be the collection of subsets such that in $[k_r] \subseteq \cup_{i\in\mathcal{I}}A_{r,i}$. By Lemma 4.4, we know that $|\mathcal{I}| = pk_r$. We have

$$
\begin{aligned}
b_r &\geq \min_{i\in\mathcal{I}}\left\{\sum_{j\in A_{r,i}}w_j\right\} \\
&= \min_{i\in\mathcal{I}}\left\{\sum_{j\in A_{r,i}}w_j 1_{j\leq t_r} + \sum_{j\in A_{r,i}}w_j 1_{t_r<j\leq k} + \sum_{j\in A_{r,i}}w_j 1_{j>k}\right\} \\
&\geq w_{k_r} + 0 + \frac{k}{nk_r}\sum_{j>k}w_j - \sqrt{k/k_r}\log(n)w_k \\
&\geq w_{k_r} + \frac{k}{nk_r}\sum_{j>k}w_j - \frac{\epsilon}{R}\sum_{j\leq k}w_j
\end{aligned}
\tag{20}
$$

The first step follows from the guarantee of QUANTILEESTIMATE (see Lemma 4.2), the third step follows from Lemma 4.5, the last step follows from the choice of parameters.

Meanwhile, we have

$$\frac{k}{nk_r}f([n]) = \frac{k}{nk_r}\sum_{j=1}^{n}w_j = \frac{k}{nk_r}\sum_{j\leq k}w_j + \frac{k}{nk_r}\sum_{j>k}w_j \leq \frac{\epsilon}{R}\sum_{j\leq k}w_j + \frac{k}{nk_r}\sum_{j\geq k}w_j \tag{21}$$

Combining Eq. (20)(21), we have proved

$$c_r = b_r - \frac{k}{nk_r}f([n]) \geq w_{k_r} - \frac{2\epsilon}{R}\sum_{j\leq k}w_r.$$

For the RHS of Eq. (7). Let $\mathcal{I}_r \subseteq [nk_r/k]$ be the top $k_r$ subsets of $\{A_{r,i}\}_{i\in[nk_r/k]}$. By Lemma 4.4, we know that there exists $i_r \in \mathcal{I}_r$, such that, either $|A_{r,i_r} \cap [t_r]| = 0$, or $j_r = A_{r,i_r} \cap [t_r]$ and $j_r \geq k_r$. Therefore, we have

$$b_r \leq \sum_{j\in A_{r,i_r}}w_j$$

$$= \sum_{j\in A_{r,i_r}}w_j\mathbf{1}_{j\leq t_r} + \sum_{j\in A_{r,i_r}}w_j\mathbf{1}_{t_r<j\leq k} + \sum_{j\in A_{r,i_r}}w_j\mathbf{1}_{j>k}$$

$$\leq w_{k_r} + \frac{\log^2(n)}{t_r}\sum_{j\leq k}w_j + \frac{k}{nk_r}\sum_{j>k}w_j + \sqrt{k/k_r}\log(n)w_k$$

$$\leq w_{k_r} + \frac{k}{nk_r}\sum_{j>k}w_j + \frac{2\epsilon}{R}\sum_{j\leq k}w_j$$

. The first step follows from the guarantee of QUANTILEESTIMATE (see Lemma 4.2), the third step follows from Lemma 4.5 and Lemma 4.6, the last step follows from the choice of parameters.

Consequently, we have

$$c_r = b_r - \frac{k}{nk_r}f([n])$$

$$\leq w_{k_r} + \frac{k}{nk_r}\sum_{j>k}w_j + \frac{2\epsilon}{R}\sum_{j\leq k}w_j - \frac{k}{nk_r}\sum_{j\geq k}w_j$$

$$\leq w_{k_r} + \frac{2\epsilon}{R}\sum_{j\leq k}w_j.$$

This completes the proof. $\qquad\square$

Now we can wrap up the proof of Theorem 4.1

*Proof of Theorem 4.1.* We prove LINEARSUM$(f, k)$ approximates the optimal value within $(1 \pm O(\log(n)\epsilon))$ factor with probability at least $4/5$, and it draws at most $(n/k) \cdot \text{poly}(n, \epsilon^{-1})$ samples.

For the correctness guarantee, we condition on the event of Lemma 4.3, which holds with probability at least $5/6$. For $r \leq R$, by Lemma 4.3, we have

$$\sum_{r=1}^{R}(k_r - k_{r-1})c_r = \sum_{j=1}^{R}w_j \pm 2\epsilon\sum_{j\leq k}w_j. \tag{22}$$

For $j \in [R+1, R+R_1]$, we have

$$\sum_{r=R+1}^{R+R_1} (k_r - k_{r-1})c_r \leq \sum_{r=R+1}^{R+R_1} (k_r - k_{r-1})w_{(1-2\epsilon)k_r} + 2\log(n)\epsilon \sum_{j \leq k} w_j$$

$$\leq \sum_{j=k_R+1}^{k_{R+R_1}} w_j + O(\log(n)\epsilon) \sum_{j \leq k} w_j \qquad (23)$$

where the first step follows from Lemma 4.3 and $k_r - k_{r-1} \leq \epsilon k_r$. Similarly, we have

$$\sum_{r=R+1}^{R+R_1} (k_r - k_{r-1})c_r \geq \sum_{r=R+1}^{R+R_1} (k_r - k_{r-1})w_{(1+2\epsilon)k_r} - 2\log(n)\epsilon \sum_{j \leq k} w_j$$

$$\geq \sum_{j=k_R+1}^{k_{R+R_1}} w_j - O(\log(n)\epsilon) \sum_{j \leq k} w_j \qquad (24)$$

Finally, for $r \in [R + R_1 + 1 : R + R_1 + R_2]$, by the guarantee of ESTIMATEQUANTILE (see Lemma 4.2)

$$\sum_{r=R+R_1+1}^{R+R_1+R_1} (k_r - k_{r-1})c_r = \sum_{j=k_{R+R_1}+1}^{k_{R+R_1+R_2}} w_j \pm O(\epsilon) \cdot \sum_{j \leq k} w_j \qquad (25)$$

Combining Eq. (22)–(25), we have

$$\sum_{r=1}^{R+R_1+R_2} (k_r - k_{r-1})c_r = \sum_{j=1}^{k} w_j \pm O(\log(n)\epsilon) \cdot \sum_{j \leq k} w_j$$

For the sample complexity, for $r \in [R + R_1]$, the total number of samples to obtain $\{b_r\}_{r \in [R+R_1]}$ is at most

$$\sum_{r=1}^{R+R_1} 100(nk_r/k)\log(n)/k_r\epsilon^2 = (n/k) \cdot \mathrm{poly}(\log(n), \epsilon^{-1}),$$

for $r \in [R + R_1 + 1 : R + R_1 + R_2]$, the total number of samples to obtain $\{b_r\}_{r \in [R+R_1+1:R+R_1+R_2]}$ is at most

$$\sum_{r=R+R_1+1}^{R+R_1+R_2} 100n\log(n)/k_r\epsilon^2 = (n/k) \cdot \mathrm{poly}(\log(n), \epsilon^{-1}).$$

This completes the proof. $\qquad \square$

Next we prove Theorem 4.7. We reduce from a decision version of the distributed set detection problem.

**Definition B.1** (Distributed index detection). Let $n, m$ be input parameters, $\mathcal{D}_0, \mathcal{D}_1$ be two Bernoulli distributions with mean $\mu_0, \mu_1$. $m$ is the number of parties, who communicate in the blackboard model. The input of the $t$-th party ($t \in [m]$) is a vector $X_t \in \{0, 1\}^n$ such that

- **YES Instance:** $X_{t,i} \sim \mathcal{D}_0$ for $i \in [n]\setminus\{i^*\}$ and $X_{t,i^*} \sim \mathcal{D}_1$;

- **NO Instance:** $X_{t,i} \sim \mathcal{D}_0$ for all $i \in [n]$

The goal is to distinguish between the YES/NO instance.

The communication complexity of distributed index detection is at least $\Omega(n/c \log(n))$, because there is an $\Omega(n/c)$ lower bound of finding the index $i^*$ in the YES instance (taking $k = 1$, $\epsilon = 1/2$ in Theorem 3.2), and one could find the index $i^*$ by performing binary search using $O(\log(n))$ calls to distributed index detection.

*Proof of Theorem 4.7.* Given $n, k, \alpha$, suppose there exists an algorithm ALG that makes at most $R$ queries and approximates the value of optimal solution. Consider an instance of distributed index detection with

$$\epsilon = \alpha/15, \qquad n' = n/k, \qquad m = \frac{10 \log(n)}{\epsilon^2}, \qquad \mu_0 = \epsilon, \qquad \mu_1 = 1/2.$$

Let $X_1, \ldots, X_m \in \{0,1\}^{n'}$ be the input of distributed index detection. Consider the following function $f : [n] \to \mathbb{R}^+$,

$$f(S) = \sum_{i \in S} f(i) \quad \text{and} \quad f(i) = \sum_{t \in [m]} X_{t, i \pmod{n'}}.$$

It is easy to see that $f$ is additive and monotone. By Lemma 3.10, in the YES instance, by taking $S^* = \{i : i = i^* \pmod{n'}\}$, we have

$$f_{\text{yes}}(S^*) \geq k f(i^*) \geq (1/2 - \epsilon) mk. \tag{26}$$

In the NO instance, for any set $S$ of size at most $k$, we have we

$$f_{\text{no}}(S) \leq 2\epsilon mk. \tag{27}$$

Let $\text{OPT} = (1/2 - \epsilon) mk$. Consider the following communication protocol: The protocol proceeds in $R$ rounds, where in the $r$-th round ($r \in [R]$), suppose ALG queries set $S_r$, then the $m$ parties collectively compute the value of $f(S_r)$. Since

$$f(S_r) = \sum_{i \in S_r} \sum_{t \in [m]} X_{t, i \pmod{n'}} = \sum_{t \in [m]} \sum_{i \in S_r} X_{t, i \pmod{n'}},$$

it suffices for the $t$-th party to compute $\sum_{i \in S_r} X_{t, i \pmod{n'}}$ locally and write it on the blackboard. Given the knowledge of $S_1, \ldots, S_r, f(S_1), \ldots, f(S_r)$, $m$ parties can simulate ALG and determine the next query $S_{r+1}$, and therefore, continue the protocol. Finally, $m$ parties could distinguish between (1) $f(S^*) \geq \text{OPT} = (1/2 - \epsilon) mk$ and (2) $f(S^*) \leq \alpha \text{OPT} = \alpha(1/2 - \epsilon) mk$, and therefore, resolve the distributed index detection task (see Eq. (26)(27)).

The communication cost at each round equals $m \cdot \log(n) = O(\log^2(n)/\epsilon^2)$, and there is a sequence of $R$ queries, so there are $O(R \log^2(n)/\epsilon^2)$ bits of communication in total. Hence, we have

$$R \log^2(n)/\epsilon^2 \geq \epsilon n'/\log(n) \quad \Rightarrow \quad R \geq \alpha^3 n/k \log^3(n).$$

$\square$

