# OpenReview forum: "A Near Linear Query Lower Bound for Submodular Maximization"
_ICML.cc/2025/Conference — ICML 2025 poster_

### Official Review · Reviewer_34cm · 2025-03-14

**Overall Recommendation:** 3

**Summary:**

The authors present query lower bounds for the problem of maximizing a monotone submodular function under a cardinality constraint.
Specifically, they improve the already existing lower bound of $\Omega(n/k)$ to $\tilde{\Omega}(n)$.
The bound holds for even estimating the value of the optimal set.

They also study the special case of additive functions (with non-negative weights).
They show that in this case one can find the optimal value with $\tilde{O}(n/k)$ queries
and that this is optimal up to polylog factors. However, they prove that finding the optimal set still requires
$\Omega(n)$ queries.

The main technique used in the paper is to reduce submodular maximization to distributed set detection.

## update after rebuttal
I keep my score.

**Claims And Evidence:**

Yes

**Essential References Not Discussed:**

None that I know of

**Experimental Designs Or Analyses:**

N/A

**Methods And Evaluation Criteria:**

Yes

**Other Comments Or Suggestions:**

The paper title in open review does not say "nearly-linear" and just says "linear". This should be corrected;
the lower bounds are not linear!
Algorithm 1: I believe this is a reduction from distributed set detection to submodular maximization, not the other way round.

**Other Strengths And Weaknesses:**

The paper studies a natural problem which is previously studied in the literature
and provides a noticeable improvement. The techniques used are interesting and may be of independent interest.
For the special case of additive functions, the results are somewhat surprising.

The main weakness is that the improvement compared to the previous work is in a somewhat limited setting.
Specifically, previous papers essentially already obtain a nearly-linear query lower bound for both very large $k$ and very small $k$.
The paper's main contribution therefore is to further study what the role of $k$ would be in such lower bounds.
While I agree that $k$ is a natural parameter, one could argue that the existing lower bounds were already "nearly-linear".

**Questions For Authors:**

See above

**Relation To Broader Scientific Literature:**

The paper is of interest to the submodular community.

**Theoretical Claims:**

I have not checked in detail but overall the claims and sketches make sense

---

> ### Author Rebuttal · Authors · 2025-04-01
>
> We thank the reviewer for the helpful feedback, we will incorporate your suggestions into our paper.

---

### Official Review · Reviewer_eFo7 · 2025-03-14

**Overall Recommendation:** 3

**Summary:**

They give a lower bound for monotone submodular maximization under cardinality constraint, showing any algorithm achieving a constant approximation factor requires $\Omega(n)$ query complexity.

They also provide an algorithm which estimates the maximum value when the function is additive using $\tilde{O}(n/k)$ query calls.

**Claims And Evidence:**

They have completely proved their claims.

**Essential References Not Discussed:**

No

**Experimental Designs Or Analyses:**

N/A

**Methods And Evaluation Criteria:**

N/A

**Other Comments Or Suggestions:**

I am not sure if the statement in the theorem at the end of the first page matches Theorem 3.9. Instead of 'no O(n)-query complexity,' shouldn't it be 'no o(n)-query complexity'? (using little-o instead of big-O)

**Other Strengths And Weaknesses:**

The lower bound result is interesting, especially while previous works provided $\Omega(n)$ lower bound for only large $k$, they show that it always holds and closes the gap between lower bound and upper bounds for this problem.

However, their algorithm for the additive case does not seem interesting since in the additive case, selecting $k$ elements with maximum value is enough and while they find a sublinear query complexity for this problem, the query complexity is not really important in this case.

**Questions For Authors:**

N/A

**Relation To Broader Scientific Literature:**

Previously, this problem had a general $\Omega(n/k)$ lower bound and also a $\Omega(n)$ lower bound when $k=\theta(n)$ but they could show that this lower bound holds for any $k$.

**Theoretical Claims:**

I only checked the proofs in the main part of the paper and not the appendix.

---

> ### Author Rebuttal · Authors · 2025-04-01
>
> We thank the reviewer for the helpful feedback. We will incorporate your suggestions into our paper and expand the discussion on the motivation behind our algorithm for additive functions.

---

### Official Review · Reviewer_9nYb · 2025-03-14

**Overall Recommendation:** 3

**Summary:**

The paper studies the submodular maximization problem under cardinality constraint $k$. They prove a new lower bound on the required query complexity for achieving a constant factor approximation ratio, which improves upon established results when $\text{polylog}{(n)} \le k \le \frac{n}{\text{polylog}{(n)}}$.

They also consider a special case of this problem where the function subject to maximization is also additive. For this problem, they prove the same lower bound and additionally propose an $(1 + \epsilon)$ approximation algorithm with only $\tilde{O}(\frac{n}{k})$ query complexity to find the value of the optimal solution.

**Claims And Evidence:**

Fine.

**Essential References Not Discussed:**

.

**Experimental Designs Or Analyses:**

This theoretical paper focuses on impossibility results, rendering experiments inapplicable.

**Methods And Evaluation Criteria:**

Fine.

**Other Comments Or Suggestions:**

line 118: vallue -> value

line 134 and 135: wrong parenthesis

**Other Strengths And Weaknesses:**

Strengths:

The paper uses novel and involved techniques.

The lower bound results are strong.

Weakness:

The writing could have had a better flow.

Proposing an algorithm for additive functions is not (well-)motivated.

Proposing an algorithm with the objective of finding value of optimal subset instead of the subset using sub-linear query time is motivated, but it's not convincing enough.

**Questions For Authors:**

.

**Relation To Broader Scientific Literature:**

.

**Theoretical Claims:**

The high-level ideas seem fine and sound.

---

> ### Author Rebuttal · Authors · 2025-04-01
>
> We thank the reviewer for the helpful feedback and will incorporate your suggestions into our paper.

---

### Official Review · Reviewer_2EUZ · 2025-03-16

**Overall Recommendation:** 4

**Summary:**

This paper studies the query complexity for monotone submodular maximization over cardinality constraint. The authors provide a nearly tight query lower bound $\tilde{\Omega}(n)$ for obtaining any constant factor approximation for any $k = o(n)$, which improves over the existing bound of $\Omega(n / k)$. This bounds holds even in the special case of a monotone modular (additive) function. They also show that this lower bound holds even for estimating the optimal value of a monotone submodular function. But for estimating the optimal value of a monotone modular function, they provide an algorithm which achieves a $( 1 \pm \epsilon)$-approximation in $\tilde{O}(n/k)$ queries, and show that this is tight up to polylog factors.

**Claims And Evidence:**

yes

**Essential References Not Discussed:**

The lower bound $\Omega(n / \log n)$ for $k = \Theta(n)$ provided in (Li et al., 2022) holds for the special case of monotone modular function too (see Theorem 4.2 therein).  This should be mentioned.

**Experimental Designs Or Analyses:**

No experiments included.

**Methods And Evaluation Criteria:**

yes

**Other Comments Or Suggestions:**

I recommend moving the discussion in Section 1.1 to the relevant sections, as in provide the high level idea of each result in the same section where you give the formal result, and only keep in this section a very high level description of the main techniques you use.

- Restate theorems/lemmas in the appendix for convenience for the reader.

- In the definition of $f_{yes}$ in Eq. (2), why not simply write this as one sum over $i \in S$?

- Specify the expected input in Algorithm 3 ($g: [m] \to \mathbb{R}^+, t > 0$). Also clarify where do you query $g$ on line 6.

- On line 412, make it clearer that you're discussing here the proof sketch of Lemma 4.3.

Typos:
- References to lines in Algorithms need to be fixed throughout.
- In Definition 3.1, $I$ should be $\mathcal{I}$.
- In Figure 1, $i$ should be $i_1$?
- In Algorithm 3, line 1 the inequality should be $>$.
- LINEARSUM is mispelled a few times on p. 7

**Other Strengths And Weaknesses:**

Already highlighted above.

Other weaknesses:
The paper lacks clarity due to numerous typos and insufficient details in the proofs

**Questions For Authors:**

1- Are there any existing algorithm for estimating the optimal value of a monotone modular function using sublinear queries?

2- Are there any existing query lower bounds for estimating the optimal value of a monotone submodular function?

**Relation To Broader Scientific Literature:**

Obtaining constant factor approximation algorithms for monotone submodular maximization with low query complexity is important for applications where function evaluations are costly, such as neural networks training and influence maximization. Estimating the optimal value is also valuable in some cases, for example to assess dataset quality before proceeding with further analysis or selection, or as a subroutine in some submodular optimization algorithms.

Existing query bounds for constant factor approximation in monotone submodular maximization are $\Omega(n / k)$ for any $k$ and $\Omega(n / \log n)$ for $k = \Theta(n)$. These bounds rule out sublinear query complexity algorithms for
$k = \Theta(1)$ and $k = \Theta(n)$. This work strengthen these impossibility results, showing that sublinear query complexity is also not possible for any $k \ll n$. This is particularly relevant as many applications involve $k$ that is not constant but still much smaller than $n$.

The lower bound  $\tilde{\Omega}(n)$ provided here holds for the special case of modular functions too. This is also the case for the existing $\Omega(n / \log n)$ bound (see Theorem 4.2 in (Li et al., 2022)) but not for the $\Omega(n / k)$ bound.
The provided lower bound holds also for estimating the optimal value of a monotone submodular function. I am not aware of other existing lower bounds for this.

Existing algorithms for submodular maximization achieve a $(1 - 1/e - \epsilon)$-approximation, using $O(n \log(1 /\epsilon))$ queries with a randomized algorithm, and $O(n/\epsilon)$ queries with a deterministic algorithm. So the $\tilde{\Omega}(n)$ query lower bound in this paper is tight up to log factors.

From a technical perspective, prior lower bounds are based on counting arguments. This paper uses novel techniques. Namely the lower bound for finding an approximate solution is derived via a reduction from the query complexity of submodular maximization to the communication complexity of a problem called distributed set detection. The lower bound for estimating the optimal value of a monotone submodular function is obtained by constructing a challenging submodular instance.

The proposed algorithm for estimating the optimal value of a monotone modular function uses two intuitive subroutines. I am not aware of existing algorithms for this problem that use sublinear queries, so this might be the first.

**Theoretical Claims:**

I checked all proofs except those of Lemmas 3.5, 4.3, 4.5, 4.6 and Theorems 4.1 and 4.7.

 The proofs I checked are correct, but have some typos and steps which are not well explained. In general, more details and intuitive explanations should be provided in the proofs.

- In the proof of Lemma 3.6, $Pr[i_1 \in \mathcal{I}]$ should be $Pr[i_1 \in \hat{\mathcal{I}} \mid V = 1]$ on line 648-649 and $Pr[i_1 \in \hat{\mathcal{I}} \mid V = 0]$ on line 652-653. I think the latter should be equal to $ |\mathcal{I} \cap \hat{\mathcal{I}}| / (n- k + 1)$ instead of  $ |\mathcal{I}| / (n- k + 1)$  (the lemma is still correct though as this is a valid upper bound).  Also, more explanation should be given for why the equalities on both these lines hold.

- In the proof of Theorem 3.2, it's not clear why the first inequality on TV holds.

- In the proof of Lemma 3.10, the second inequality bounds the probability that $\sum_t X_{t, i}$ deviates from its mean by more than $\epsilon m/2$, while the Lemma requires bounding the probability of $\sum_t X_{t, i}$ not being in $(\epsilon m/2, 2 \epsilon m)$, so the bound given doesn't actually match that. The lemma is still correct, but the proof needs to corrected.

- In the proof of Lemma 4.4, give more details for why the first two inequalities hold (1st one can be proved by induction, for 2nd one remind the reader of how all these quantities are related..). Moreover why is Azuma-Hoeffding bound needed, this bound holds by Hoeffding inequality.

---

> ### Author Rebuttal · Authors · 2025-04-01
>
> We thank the reviewer for the constructive feedback and will incorporate your suggestions into our paper.
>
> > Are there any existing algorithms for estimating the optimal value of a monotone modular function using sublinear queries?
>
> To the best of our knowledge, no such algorithms currently exist.
>
> > Are there any existing query lower bounds for estimating the optimal value of a monotone submodular function?
>
> Yes—the $\Omega(n/k)$ lower bound established in prior work also applies to the estimation of the optimal value.

---

### Decision · Program_Chairs · 2025-05-01

**Decision:**

Accept (poster)

**Comment:**

The paper studies the query complexity of monotone submodular maximization with a cardinality constraint and provides improved lower bounds on the query complexity for achieving a constant factor approximation. Prior work has shown a lower bound of  $\Omega(n/k)$ and this work strengthens this result to a nearly-tight $\widetilde{\Omega}(n)$ lower bound. For additive functions, the paper gives a sublinear algorithm for estimating the value of the optimal solution.

Overall, this work is a valuable addition to the submodular maximization literature and it is relevant to the community. The reviewers appreciated the main contribution of this work and support accepting the paper. There was consensus among the reviewers that the main result is strong and the techniques are novel.